# Nursing Students’ Subjective Happiness: A Social Network Analysis

**DOI:** 10.3390/ijerph182111612

**Published:** 2021-11-04

**Authors:** Eun-Joo Kim, Ji-Young Lim, Geun-Myun Kim, Seong-Kwang Kim

**Affiliations:** 1Department of Nursing, Gangneung-Wonju National University, Wonju 26403, Korea; kimeju@gwnu.ac.kr (E.-J.K.); tjdrhkd141@gwnu.ac.kr (S.-K.K.); 2Department of Nursing, Inha University, Incheon 22212, Korea; lim20712@inha.ac.kr

**Keywords:** nursing students, subjective happiness, social network analysis

## Abstract

Improving nursing students’ subjective happiness is germane for efficiency in the nursing profession. This study examined the subjective happiness of nursing students by applying social network analysis (SNA) and developing a strategy to improve the subjective happiness of nursing. The study adopted a cross sectional survey to measure subjective happiness and social network of 222 nursing students. The results revealed that the centralization index, which is a measure of intragroup interactions from the perspective of an entire network, was higher in the senior year compared with the junior year. Additionally, the indegree, outdegree, and centrality of the social network of students with a high level of subjective happiness were all found to be high. This result suggests that subjective happiness is not just an individual’s psychological perception, but can also be expressed more deeply depending on the subject’s social relationships. Based on the study’s results, to strengthen self-efficacy and resilience, it is necessary to utilize strategies that activate group dynamics, such as team activities, to improve subjective happiness. The findings can serve as basic data for future research focused on improving nursing students’ subjective happiness by consolidating team-learning social networks through a standardized program approach within a curriculum or extracurricular programs.

## 1. Introduction

University is an important developmental period in an individual’s life, characterized by the onset of various stress-induced mental problems [1]. It is also a period of academic and developmental difficulties, as demonstrated by the mental health issues among college students worldwide; for example, American college students have a 7% prevalence of anxiety disorders, which is higher than the population average by 4.3–5.9% [2]. College students’ subjective happiness has thus become a major research area [3]. Subjective well-being, or perceived mental health status, is defined as an evaluative response to personal life. Further, it consists of cognitive factors, such as life satisfaction, and emotional factors, such as happiness [3]. It is associated with healthy behaviors, sound mental health, and improved work performance and social and personal skills [4]. Poor mental health in college students can result, for example, from smoking, substance abuse, poor academic performance, and dropout [3,5]. Most studies on college students’ subjective happiness conducted to date have examined variables related to their psychological resources, such as self-efficacy, resilience, and positive personality characteristics [6,7,8,9]. Subjective happiness refers to a cognitive appraisal of life satisfaction based on individual experience viewed from one’s own perspective [10]. High subjective happiness protects individuals from crises by mitigating the harmful effects of negative emotions on concentration, creativity, and other aspects [11]. Going a step further enhances psychological well-being, elevating the inner state to the optimal level.

As nursing has the goal of improving the patient’s health and quality of life to ultimately promote happiness, helping nursing students to seriously consider their subjective perceptions of a happy life and build their own values for happiness can serve as an opportunity to foster nursing students’ care skills and sensitivity to patients [12]. Further, the subjective happiness of nursing students is significant as it contributes to their mental well-being. This helps create a positive and vibrant nursing environment where the nurse has a high level of happiness and an in-depth understanding of the patient’s happiness [13]. The study of subjective happiness in nursing students includes studies of happiness perception and attempts to comprehensively identify various individual psychological and social factors that affect happiness. In particular, it includes studies on satisfaction with one’s academic specialty, stress, self-efficacy, self-respect, self-resilience, and social support [12,14,15,16]. It has been noted that nursing students’ subjective happiness helps them adapt better to college life by attaching importance to their role in helping others and gaining a more positive self-image [17]. Moreover, it has been reported that nursing students’ increased subjective happiness during their college years could lead to high-quality nursing care [18].

Nevertheless, nursing students are reported to have four times the stress of those in other academic specialties, as well as lower levels of life satisfaction and subjective happiness [19]. Nursing students experience other stressors that their peers in other degree programs do not, such as those related to their academic program and their role as nursing students [20]. Furthermore, nursing students have more professional experiences during their studies than students in other programs, which adds stressors related to responsibilities for their patients’ well-being, making it harder for them to participate in student campus life and normal social activities [21]. In particular, Korean nursing students were found to have a low sense of happiness during clinical practice due to the burden of nursing patients, lack of knowledge and experience, and strict norms [14,15].

Happiness is a subjective and mental concept of an individual in addition to being a social and cultural concept [22]. Studies define happiness as determined by an individual’s temperament, personality, and experience and have found that it is achieved through social relationships and leisure activities [23,24].

Nurses must have clinical performance and communication skills based on their knowledge of their specialty as well as collaborative skills to work with others, including fellow nurses and other medical professionals. As these are recognized as important abilities, Korean nursing students must learn these skills in their university courses. Thus, various team-based teaching methods are used in both clinical practice and theory courses; they are also used to enhance students’ abilities to provide holistic care to patients through cooperation with multidisciplinary teams during their future nursing practice at clinical sites. Therefore, team learning is considered an important teaching method at nursing colleges. However, due to a lack of consideration for others, communication problems, and passive participation in the team-based learning process, the burden of team-based learning is becoming overwhelming. Negative perceptions have also led to reports of poor satisfaction and effectiveness, leading to questions about the method [25]. Clinical practicums in nursing education emphasize team learning, so their social interactions are significant.

The term “social network” refers to a network of connections among people linked by a network formed naturally through interactions among individual members from different fields of expertise [26]. The social network theory attempts to explain human actions and social structures, focusing on relational ties, wherein positions of the actors within a particular form of the network affect their perceptions, values, and rewards for their actions [27]. Based on this theory of social networks, analysis is performed on the actions arising from interactions within the network, not on individual attributes, by analyzing the forms of social relationships or patterns of social connections [28]. A social network helps its members enhance team-based cooperative learning outcomes through its resources and knowledge channels and experience positive learning effects and group dynamics [29,30]. According to this approach, human actions are determined by the network positions and shapes, whereby the network keeps renewing itself through interactions among its members, which in turn influence further human actions [31]. This means that interpersonal and intragroup relationships and individual behaviors and achievements can be explained by analyzing social networks [32].

The social network serves as a channel of resource and knowledge exchange among students, enhancing the efficacy of cooperative learning and the resultant academic achievement [29,30]. Therefore, when considering nursing students’ characteristics, it is worth examining the dynamics in social relationships, using a structural perspective of social network analysis (SNA) to explore the relationship between college students’ subjective happiness and the variables representing emotional characteristics.

### 1.1. Purpose of the Present Research

This study was conducted to identify the levels of subjective happiness in nursing students, depending on their social networks. To this end, the following objectives were set:(1)Identifying nursing students’ intragroup interactions and sociogram;(2)Identifying the correlation between subjective happiness and the variables of SNA.

### 1.2. Theoretical Thesis

As mentioned, happiness is both an individual’s subjective and mental concept and a social and cultural concept [22]. Bruner [33] argued that, as meaning and concepts are formed by culture, people’s conception of happiness is also a concept formed in the context of a unique culture. For this reason, various factors that affect happiness will be different depending on cultural characteristics. Most theories about happiness argue that happiness is determined by individual personality, experience, and temperament [23] and that happiness is achieved through social relationships, leisure activities, and various other activities [24]. However, Chinese students describe a conceptual framework differently from Western students [34]. Uchida and Oishi [35] believe that people in oriental cultures are happy when they feel secure within limits accepted by society. Conversely, in Western culture, individual needs and emotions are more important, and social limitations are thought to be restrictions on individuals. The current study identifies the relationship of Korean nursing students’ subjective happiness to social network elements, such as centrality, indegree, and outdegree. It determines if there are differences between individuals.

## 2. Materials and Methods

### 2.1. Study Design

This study used a cross-sectional, qualitative design utilizing a questionnaire to measure subjective happiness and social network support.

### 2.2. Participants

The participants in this study were third- and fourth-year nursing students at two universities in a medium-sized city in South Korea. Nursing department A had 75 students per grade, totaling 300 students, and the lectures were conducted in two sections. Nursing department B had 50 students per grade with a total of 200 students. The lectures by grade were operated in one class. Except for cases where the size of the nursing department was too large due to the characteristics of SNA, schools with a total of 50–100 students per grade were included using convenience sampling. The boundaries of the populations studied by social network analysts are of two main types. Probably most commonly, the boundaries are those imposed or created by the actors themselves. All the members of a classroom, organization, club, neighborhood, or community can constitute a population. These are naturally occurring clusters, or networks. Therefore, in a sense, social network studies often draw the boundaries around a population that is known, a priori, to be a network [36]. In this study, two nursing colleges and two grades were included, and the number was 250.

This study was conducted only for third- and fourth-year students who had clinical practice experience considering the characteristics of the nursing students. The research was conducted from 1 September 2019, to 31 October 2019. The purpose and method of this research were explained in a meeting with the nursing department director. Further, the research was conducted after obtaining the participants’ consent. Questionnaires were distributed to a total of 250 students. All 250 people responded to the questionnaire, and 17 people (6.8%) who did not fill out such as monthly allowance or friendship were excluded. Finally, 233 (93.2%) questionnaires were used for the analysis. The final sample comprised 68 third-year students (A3) and 78 fourth-year students (A4) from department A and 45 third-year students (B3) and 42 fourth-year students (B4) from department B. The average age (standard deviation) of the participants was 21.72 years (1.44), 23.23 years (2.43), 21.62 years (1.37), and 22.52 (1.17) in A3, A4, B3, and B4, respectively. Women comprised 79.4% of the A3 classes, 83.3% of A4, 73.3% of B3, and 78.6% B4.

### 2.3. Measures

#### 2.3.1. Subjective Happiness

Subjective happiness was measured using the Oxford Happiness Questionnaire [37] used in [38] for undergraduate students. The scale consists of 29 items rated on a five-point Likert scale ranging from 1 (strongly disagree) to 5 (strongly agree). A higher score indicates a higher subjective happiness level. The Cronbach α in Robbins et al. [38] was 0.91, and the Cronbach α in the present study was 0.91.

#### 2.3.2. Social Network

Social network data can be measured via in-person interviews or self-administered questionnaires, and it is essential to clearly set forth the scope and boundary of the study target [39]. SNA is expressed by an actor, node, and relationship (tie) or link. Indicators that characterize the network structure include the number of nodes and links: network size, network density, degree of connectivity, centrality, distance, average distance, diameter, and reciprocity. The degree of connectivity is the most basic indicator used to describe the characteristics of each node within the network, expressed in terms of the number of nodes directly connected to each node (or the number of connecting lines). It is recognized as an indicator of the influence or activity of a particular node [40]. The more central it is in the network, the more influential it is in the group, and the easier it is to obtain information. The centrality index, which is widely used in SNA, is central to the degree of connectivity, expressed largely in terms of “indegree centrality” and “outdegree centrality” [41]. This is the number of connections each node in the network receives from other nodes and gains popularity and leadership. Outgoing centroid, in contrast, means the number of connections each node gives to another node [42]. In this study, social networks were measured using the “peer nomination” method in which each participant is instructed to give the names of three to seven friends, which is the number required for network construction. The items for the peer network questionnaire were taken from the items proposed by Bell [43].

### 2.4. Procedure

Inquiries were made to the heads of the nursing departments of two universities in the same region in South Korea for the recruitment of study participants. After explaining to the director that the entire sampling unit should be included, considering the nature of SNA, a research assistant was allowed to explain the research objectives and procedures to the junior and senior classes. After explaining to the participants that all the data would be coded, the autonomy and confidentiality of participation would be assured, and withdrawal at any point would entail no disadvantage, the research assistant distributed the questionnaire to those who submitted written consent. The participants were given the questionnaire and a list of the same-year students and were asked to record the names of the peers they felt close to as part of the study. They were instructed to submit the completed questionnaire in a sealed envelope to prevent content from being accidentally viewed. We offered participants a souvenir as a token of appreciation.

### 2.5. Data Analysis

We processed the collected data using Gephi and SPSS 24.0.

(1)General characteristics and subjective happiness of nursing college students were analyzed by frequency, average, and standard deviation using SPSS 24.0.(2)The analysis was conducted using Gephi to express and measure the sociogram among nursing college students visually. The existence and absence of relationships within the network were entered as 1 and 0, respectively. In this study, analysis was conducted in a directed mode that was a complete network and was used when the properties of rows and columns were equal in “person × person”. The layout was visualized by specifying a group and comparing network types by the group. Indegree, outdegree, between centrality ties, density, geodesic distances, clustering coefficient, and reciprocity were identified to assess network characteristics between the group.(3)The correlations between subjective happiness, indegree, outdegree, and centrality were analyzed using Pearson’s correlation coefficients.

### 2.6. Ethical Considerations

All procedures performed in studies involving human participants were per the ethical standards of the institutional or national research committee and with the 1964 Helsinki Declaration and its later amendments or comparable ethical standards. Before data collection, the Institutional Review Board (IRB No. ***** 2019-24) of the university to which the researcher belonged approved this study. Written consent was obtained from all individual participants included in the study. As students in the university to which the researcher belonged can be considered vulnerable (i.e., the researcher’s students), a questionnaire was distributed by students and a co-researcher who did not know the students. In addition, personal identification information was coded to ensure anonymity, and the students were told data would not be used for purposes other than those explained in the research.

## 3. Results

### 3.1. Participants’ General Characteristics

In all four groups, female students outnumbered male students, accounting for more than 70.0% of the sample. In terms of the participants’ religion, no religion and Buddhism were the most and least frequent answers in all four groups, respectively. Self-sustaining was the most common residential type in all four groups, and the parental home or dormitory was the least common. The four age groups and allowance showed a statistically significant difference (Table 1).

### 3.2. Network Interactions and Sociograms of Each Group

The degree of connection is an indicator of how many connection relationships a node has in a network. As the number of links connected to a node increases, the centrality of the degree of connection of the node increases. Indegree is the degree to which one node is selected by other nodes. Outdegree is the degree to which one node pays attention to other nodes. Therefore, indegree is also expressed as popularity, and outdegree is also expressed as activity. Mediation centrality is an indicator of how much a particular node acts as an intermediary between other nodes in a network. Nodes with high mediation centrality secure high control over the flow of information and resources within the network [39,40,41].

Figure 1 shows the sociogram of each group. The number of nodes reflects the number of actors in the network, and ties refers to the number of connected nodes, which represents the interdependent relationship between nodes. Density is a numerical representation of how many relationships all members have with each other within the network. Geodesic distance is the shortest distance connecting two nodes, and the smaller this value is, the faster the transition speed between members. The clustering coefficient is a value indicating how tightly nodes are clustered within the network. Reciprocity is a value indicating the interaction between whether the network acts in one direction or both directions.

In all figures, the size of the node is the variable value of each social network, and the color of the node means the subjective happiness average. The redder the color, the higher the feeling of happiness, and the larger the size of the node, the larger the variable value.

In the sociogram of A3, the shapes of indegree, outdegree, and between centrality were all similar, the size of the node was similar to that of indegree, and between centrality was significantly small (Figure 1).

The numbers of nodes and ties for A3 were 68 and 221, respectively, geodesic distance was 4.11, and density was 0.049. There was one cluster separated from the overall network and two isolated nodes. The overall clustering coefficient was 0.47. The connectivity of nodes positioned at the center of each cluster was high, but no specific hub of nodes was observed. This can also be confirmed by the difference between the overall clustering coefficient and the weighted overall clustering coefficient (0.009). The greater the difference between these two values, the greater the degree of inequality among the network size of individual nodes. A3′s reciprocity was high at 0.73 (Figure 1).

An analysis of inter-variable correlations of the A3 group revealed that subjective happiness had small but positive correlations with indegree (r = 0.19, *p =* 0.115), outdegree (r = 0.28, *p* = 0.019), and between centrality (r = 0.08, *p =* 0.501). Statistically, the correlation between subjective happiness and the outdegree of A3 was significant and similar to the shape of the sociograms (Table 2).

The numbers of nodes and ties for A4 were 78 and 279, respectively, with its sociogram showing a densely meshed net. The geodesic distance and density were 4.45 and 0.05, respectively. There was a cluster separated with all links cut in the top right corner and an isolated node. With the clustering coefficient of 0.49, many clusters were found across the network. Further, no hub was observed around which the network was formed. A4 had the highest number of links (0.279) and high reciprocity (0.789). However, a completely separated cluster appeared in the top right corner. The difference between the overall and weighted clustering coefficients was 0.049, demonstrating a high degree of inequality in network size. A4′s betweenness centrality, at 9.5, had the lowest value among all groups, which indicates a small number of nodes playing the bridging role.

When examining the plot of A4 where node color represents subjective happiness (red = very happy, orange = happy, and white = very unhappy) and the size of each node represents indegree, we find all but nine nodes are small, regardless of their color. An analysis of inter-variable correlations of the A4 group revealed that subjective happiness had small but positive correlations with indegree (r = 0.32, *p =* 0.004), outdegree (r = 0.17, *p =* 0.137), and between centrality (r = 0.18, *p* = 0.109). Statistically, the correlation between subjective happiness and indegree of A4 was significant and similar to the shape of the sociograms (Table 2).

The nodes and ties for B3 were 45 and 195, respectively, with its sociogram showing a large triangular shape with distinct clusters. The geodesic distance and density were 3.44 and 0.10, respectively. No separate clusters or isolated nodes were observed, and with 4.33, the average degree had the highest value of all groups. With the difference between the overall and weighted clustering coefficients calculated at 0.054, B3 showed the highest degree of inequality in network size. This can also be confirmed in the sociogram, as revealed by comparing the bottom left and right clusters. Likewise, looking at the clusters in the center and the top right corner, connections were formed around particular nodes. The relatively high betweenness centrality (12.96) suggests that many nodes play the role of mediating clusters. This type of network, that is, high inequality in connectivity between nodes and high betweenness centrality, suggests that clusters are prone to complete separation when the connection to the mediating node is cut.

In the sociogram of B3, compared to other groups, the size of the node was larger and the size of the orange nodes was significantly larger. In addition, subgroups with similar node sizes and colors were noticeable (Figure 1).

An analysis of inter-variable correlations of the B3 group revealed that subjective happiness had small but positive correlations with indegree (r = 0.31, *p* = 0.035), outdegree (r = 0.38, *p* = 0.011), and between centrality (r = 0.19, *p* = 0.207). Statistically, the correlation between subjective happiness and indegree and outdegree of B3 was significant and similar to the shape of the sociograms (Table 2).

The geodesic distance and density for B4 were 4.587 and 0.08, respectively. Despite having the smallest group size, its paths showed the highest number of node levels. Its spider web-like sociogram shows the most clearly defined clusters of all groups. The clusters are evenly distributed throughout the network, which can also be confirmed by the difference between the overall and weighted clustering coefficients as low as 0.004. However, with the lowest reciprocity (0.696), B4 had the highest number of unilateral links, although it had the smallest links. Conversely, its high betweenness centrality at 30.21 suggests that there are a large number of nodes forming indirect relationships with each other. In this type of network, separated clusters or isolated nodes are rarely found.

An analysis of inter-variable correlations of the B4 group revealed that subjective happiness had small but positive correlations with indegree (r = 0.40, *p* = 0.008), outdegree (r = 0.40, *p* = 0.008), and between centrality (r = 0.16, *p* = 0.312). Statistically, the correlation between subjective happiness and indegree and outdegree of B4 was significant and similar to the shape of the sociograms (Table 2).

## 4. Discussion

Due to the specific nature of the two-track curriculum consisting of theoretical learning and clinical practicum, nursing education emphasizes team learning. The key factor for successful team learning is smooth interactions among team members [43]. Such interactions could be assessed by analyzing the attributes of social relationships, which SNA can achieve [44]. Moreover, the social network serves as a channel of resource and knowledge exchange among team members, enhancing the efficacy of cooperative learning and the resultant academic achievements [29,30]. According to Benton et al. [45], social network analysis studies on nursing have increased rapidly over the past 20 years, contributing to expanding insights from a nursing perspective, such as identifying interaction patterns. Cho et al. [29] found that social networks are very useful in cultivating the necessary competencies for jobs that require collaboration with various groups, such as nursing, by forming a channel for mutual exchange of knowledge and resources. Further, Rabbany et al. [46] emphasized that SNA can be a useful tool for fair evaluation of student participation in the increasing online curriculum.

In this study, the centralization index, which is a measure of intragroup interactions from the perspective of an entire network, was higher among fourth-year students as compared to third-year students. In particular, A3 showed the lowest centralization index of all groups, and B4 had by far the highest centralization index. As demonstrated in Figure 1, A3 shows a linear network close to a straight line, with a portion of the cluster severed from the main network and isolated. This suggests that the interactions among the students linearly take place centering on a few specific members, resulting in a weak degree of interactions and weak connectivity. Conversely, B4′s network shape is close to a sphere with lively links and exchanges among all members without forming a particular hub attracting traffics. This suggests that no particular member monopolizes the central role, but all members interact, forming an organic whole, generating lively information and opinion exchanges.

A high centralization index means that there are concentrated exchanges among the participants with high connectivity. Conversely, a low centralization index points to a wide variety of exchange patterns among the participants with an extended network with weak but broad connectivity [6]. Piao and Kim [44] performed an SNA of learner interactions in nursing students. They reported a trend toward centralization of interactions among the members in later years of university, which is consistent with the results of our study.

The finding of this study, namely that nursing students with high subjective happiness scores showed a high degree of centrality within a social network, can be explained by their ability to form and maintain harmonious interpersonal relationships with the people around them [47,48]. This happiness score is also associated with high indegree centrality, which facilitates the reception of learning information and resources from other members and the positive emotions they experience when helping others [17]. Additionally, this happiness score is associated with a high outdegree centrality, which occurs through the active transmission and exchanges with other members within the network. In this study, we found that subjective happiness is an important psychological resource, positively activating nursing students’ social network and ultimately contributing to achieving learning goals and enhancing major satisfaction by mediating the positive experience of team-based cooperative learning.

The nursing-specific curriculum may explain an increase in the centralization index among seniors. That is to say, the junior year is the first year where nursing students start clinical practicum; thus, they have fewer team learning opportunities than seniors. This is as they have just begun to be engaged in team-based cooperative learning. In contrast, clinical practicum in the senior year requires a comprehensive application of all knowledge learned in theoretical lectures. Therefore, students can carry out more mature cooperative learning based on the team learning experience accumulated during the junior year. Aviv et al. [47] reported that a higher centralization index in intragroup interactions indicates a higher learning performance. Currently, nursing education in Korea is standardized as a performance-oriented curriculum according to the guidelines set out by the Korea Accreditation Board of Nursing, which makes students’ interactions and exchanges indispensable. Apart from this, nursing students can greatly benefit from the experience of team activities accumulated throughout the undergraduate years when carrying out their roles as team nurses in the healthcare facilities where they would work after graduation [18].

Independent of the positive effects of improved interactions among students, including exchanges of information and opinions and improvement in joint undertakings, there are also negative effects such as conflicts related to role allocation and communication problems [48]. Park and Choi [49] reported that overconcentration of intragroup exchanges on a few group members breaks the network equilibrium and biases information sharing, resulting in decreased overall team performance. The importance of building a mature pattern of cooperative relationships is highlighted as more skilled team interactions are required for solving complex problems going far beyond simple information sharing as college years progress [50]. To this end, it is necessary to increase the level of influence of students positioned at the network center to reduce the negative experiences marginalized students have in the network’s periphery. This leads to increased synergy among all members, constructing a harmonious social network, and achieving the team goal.

Subjective happiness is a cognitive appraisal of life satisfaction based on individual experience viewed from one’s perspective [10]. An individual with high subjective happiness can better manage crisis as they can better protect themselves from negative emotions’ harmful effects on concentration, creativity, and other merits [11]. Further steps enhance psychological well-being, elevating the inner state to the optimal level. Garlan et al. [50] reported that the positive effects of subjective happiness alleviate the detrimental psychological state in patients with mental disorders by enhancing their resilience. Park [17] showed that nursing students’ role of helping others increases their positive self-image, which boosts their subjective happiness, resulting in their better adaptation to college life.

In finding ways to enhance the subjective happiness of nursing students, this study identified the relationship of subjective happiness with social network elements, such as indegree, outdegree, and between centrality. Previous studies attempting to develop measures that could enhance subjective happiness either emphasized individual factors, as subjective happiness is an interpretation of experience based on individual perspectives, or highlighted the collective relational elements of an integrated approach to the individual’s psychological and social factors. Studies highlighting individual factors considered subjective happiness a highly personal experience with subjective awareness and perception and sometimes identified it with subjective well-being [51]. Conversely, studies highlighting collective relational aspects emphasized social aspects such as economic well-being, culture, education, and leisure, suggesting that happiness is determined by how well individuals function in their societies rather than by individual factors [52]. A study on nursing students’ perception of happiness reported that happiness, self-identity, and belonging are enhanced when receiving support from family members in the form of love, support, bonding, and encouragement [12]. Additionally, happiness depends on external standards such as human relationships, and in particular, nursing students highlighted the development of good relationships with colleagues, thus emphasizing the connectedness of human relations.

In interpreting the results of this study, care should be taken when generalizing the results, as they were generated from a small sample of Korean nursing students. Therefore, it is necessary to repeat the study targeting nursing students in various countries. For example, using a global dataset, Kondakci et al. [53] analyzed international student mobility using SNA and determined various regional hubs existed that comprised UK, French, German, and Turkish communities. As such, further SNA is needed to identify differences and commonalities between countries regarding subjective happiness of nursing students, which can then be used to improve student education.

## 5. Conclusions

In this study, students with a high level of subjective happiness showed high indegree, outdegree, and social network centrality. In particular, this phenomenon was more conspicuous for fourth-year than third-year students. This means that a high level of subjective happiness is associated with a strong social network. Furthermore, this suggests that subjective happiness is not just an individual’s psychological state or perception but can be expressed more deeply depending on the social relationship to which the subject belongs. Therefore, it is necessary to utilize strategies to strengthen self-esteem, self-efficacy, and resilience by activating group dynamics such as team activities to improve the subjective happiness of nursing students.

## Figures and Tables

**Figure 1 ijerph-18-11612-f001:**
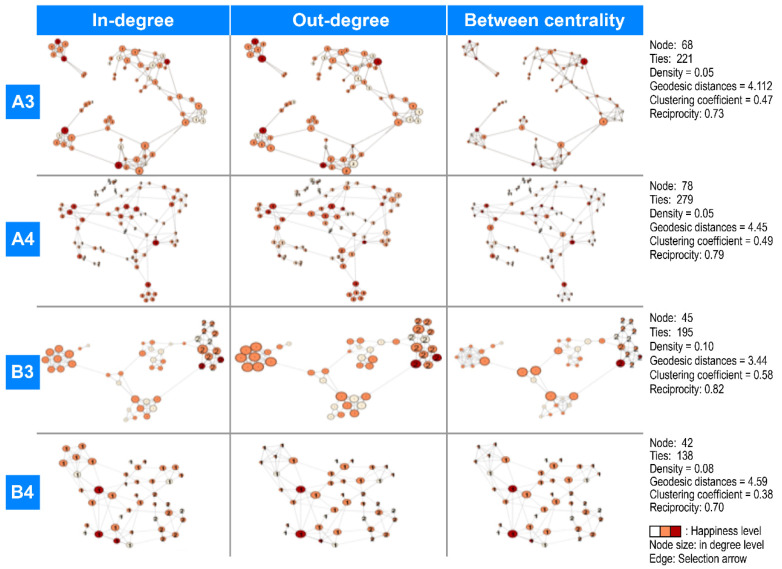
The sociogram of each group. Node color: red = very happy, orange = happy, and white = very unhappy. A3: Nodes: 68; Ties: 221; Density: 0.05; Geodesic distances: 4.112; Clustering coefficient: 0.47; and Reciprocity: 0.73. A4: Nodes: 78; Ties: 279; Density: 0.05; Geodesic distances: 4.45; Clustering coefficient: 0.49; and Reciprocity: 0.79. B3: Nodes: 45; Ties: 195; Density: 0.10; Geodesic distances: 3.44; Clustering coefficient: 0.58; and Reciprocity: 0.82. B4: Nodes: 42; Ties: 138; Density: 0.08; Geodesic distances: 4.59; Clustering coefficient: 0.38; and Reciprocity: 0.70.

**Table 1 ijerph-18-11612-t001:** Summary of the participants’ general characteristics and research variables.

General Characteristics		A3 (*n* = 68)		A4 (*n* = 78)		B3 (*n* = 45)		B4 (*n* = 42)		F or X^2^ (*p*)
		*n* (%) or M ± SD								
Age (years)		21.72 ± 1.44 ^a^		23.23 ± 2.43 ^b^		21.62 ± 1.37 ^c^		22.52 ± 1.17		11.70 (< 0.001) a,c < b **
Gender	Female	54	79.4	65	83.3	33	73.3	33	78.6	1.76 (0.624) *
	Male	14	20.6	13	16.7	12	26.7	9	21.4	
Religion	Protestant	17	25	26	33.3	11	24.4	10	23.8	
	Catholic	7	10.3	5	6.4	7	15.6	7	16.7	0.41 (0.937) *
	Buddhism	4	5.9	3	3.8	3	6.7	2	4.8	
	No religion	40	58.8	44	56.4	24	53.3	23	54.8	
Residential type	Parental home	22	32.4	17	21.8	9	20	13	31	
	Dormitory	20	29.4	20	25.6	15	33.3	4	9.5	4.04 (0.257) *
	Self-sustaining	26	38.2	41	52.6	21	46.7	25	59.5	
Allowance (Unit: 1000 won)		346.2 ± 182.8 ^a^		397.1 ± 186.7		477.8 ± 185.7 ^b^		430.5 ± 127.4		5.43 (0.001) a < b *
Subjective happiness		3.21 ± 0.33		3.20 ± 0.26		3.18 ± 0.37		3.22 ± 0.37		0.13 (0.944)

* Kruskal–Wallis test; ** Scheffé’s test. The letters ^a^, ^b^, and ^c^ in the table indicate three different groups in the Scheffé’s test.

**Table 2 ijerph-18-11612-t002:** Correlations between the research variables.

	Variables	Subjective Happiness	Indegree	Outdegree	Between Centrality
	Subjective happiness	1			
Total	Indegree	0.28 **	1		
	Outdegree	0.27 **	0.58 **	1	
	Between centrality	0.14 *	0.35 **	0.40 **	1
	Subjective happiness	1			
A3	Indegree	0.19	1		
	Outdegree	0.28 *	0.49 **	1	
	Between centrality	0.08	0.38 **	0.40 **	1
	Subjective happiness	1			
A4	Indegree	0.32 **	1		
	Outdegree	0.17	0.59 **	1	
	Between centrality	0.18	0.49 **	0.54 **	1
	Subjective happiness	1			
B3	Indegree	0.31 *	1		
	Outdegree	0.38 **	0.65 **	1	
	Between centrality	0.19	0.25	0.26	1
	Subjective happiness	1			
B4	Indegree	0.40 **	1		
	Outdegree	0.40 **	0.53 **	1	
	Between centrality	0.16	0.46 **	0.60 **	1

* *p* < 0.05, ** *p* < 0.001.

## Data Availability

The research data can be requested from the first author.

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
