# Peer review of "Nursing Students’ Subjective Happiness: A Social Network Analysis"

_ijerph, 2021, doi:10.3390/ijerph182111612_

Round 1
Reviewer 1 Report
Thank you for giving me an opportunity to review good research. This study is a valuable study to promote subject happiness of nursing students and furthermore to increase their nursing care quality for patients. Kindly find some of the issues noted below that I feel some need attention. Thanks.
Abstarct
In, conclusion, the current suggestion is considered to be a suggestion that can be considered even without the results of this study. Therefore, please describe a realistic suggestion based on the results of this study.
Introduction
“students’ subjective happiness has thus become a major research area.”: reference need.
Method
-Please add a rationale for calculating the sample size
-It is questionable whether the subjects of this study are students of the researcher. If so, these study subjects are considered vulnerable subjects. Please further describe your efforts to ensure their right to voluntary consent, etc.
Results
Place footnotes for A3, A4, B3, B4 at the bottom of the tables and figures.
Discussion
- Interpretations and suggestions for the results described by the current researchers are considered to be rather localized interpretations based on the situation in your country, Korea. Also include a discussion of the international nursing education field.
- Please refrain from repeating the same content as the introduction.
Ex: “~increased nursing students’ subjective happiness during their college years could lead to high-quality nursing care [18].”
With Thanks
Author Response
Thank you for your review.
Responses to the review comments are attached as a file.

Reviewer 2 Report
The authors present an interesting study on subjective happiness in nursing students and social networks.It is an interesting and novel approach to the topic, developing a strategy for improvement, and with respect to the introduction, it would be advisable to include more literature on aspects that influence stress and unhappiness in these students. For example, it would be interesting to review and refer to this article on stressors and coping strategies:
Anxiety, perceived stress and coping strategies in nursing students: a cross-sectional, correlational, descriptive study.
BMC Med Educ. 2020 Oct 19;20(1):370. doi: 10.1186/s12909-020-02294-z.PMID: 33081751
The number of participants in relation to the total number of invitees should be expressed as a percentage and reference should be made in the article to possible reasons for non-participation.
The Oxford Happiness Questionnaire is not a specific instrument for nursing students. This should be further explored, previous studies in this type of students if available, or a more complete psychometric analysis in this population.
Figure 1 should be adapted to the standards of the journal and authors are advised to elaborate further in the text of the manuscript to facilitate the reader's understanding.
The discussion should be revised, presented in a more orderly and concise way by discussing the main results. It would also be interesting to elaborate more on the limitations and a main conclusion.
Author Response
Thank you for your review points.
Responses to the review comments are attached as a file.
